# Breaking Barriers: The Promise and Challenges of Immune Checkpoint Inhibitors in Triple-Negative Breast Cancer

**DOI:** 10.3390/biomedicines12020369

**Published:** 2024-02-05

**Authors:** Sawsan Sudqi Said, Wisam Nabeel Ibrahim

**Affiliations:** Department of Biomedical Sciences, College of Health Sciences, QU Health, Qatar University, Doha P.O. Box 2713, Qatar; s.said@qu.edu.qa

**Keywords:** resistance, triple-negative breast cancer, PD-1/PD-L1/CTLA-4, TME

## Abstract

Triple-negative breast cancer (TNBC) is a highly aggressive malignancy with pronounced immunogenicity, exhibiting rapid proliferation and immune cell infiltration into the tumor microenvironment. TNBC’s heterogeneity poses challenges to immunological treatments, inducing resistance mechanisms in the tumor microenvironment. Therapeutic modalities, including immune checkpoint inhibitors (ICIs) targeting PD-1, PD-L1, and CTLA-4, are explored in preclinical and clinical trials. Promising results emerge from combining ICIs with anti-TGF-β and VISTA, hindering TNBC tumor growth. TNBC cells employ complex evasion strategies involving interactions with stromal and immune cells, suppressing immune recognition through various cytokines, chemokines, and metabolites. The recent focus on unraveling humoral and cellular components aims to disrupt cancer crosstalk within the tumor microenvironment. This review identifies TNBC’s latest resistance mechanisms, exploring potential targets for clinical trials to overcome immune checkpoint resistance and enhance patient survival rates.

## 1. Introduction

Breast cancer has emerged as the most prevalent cancer among women globally and ranks as the second leading cause of cancer-related mortality. Its clinical manifestations exhibit significant heterogeneity, resulting in variable treatment responses across patients [1]. The classification of breast cancer encompasses four intrinsic molecular subtypes: Luminal A (expressing progesterone and estrogen receptors), Luminal B (displaying variable proliferation and a lack of human epidermal growth factor receptor 2 (HER2), while still expressing progesterone and estrogen receptors), HER2-overexpressing (HER2+), and basal-like breast cancer (expressing genes of normal breast basal and/myoepitelial cells) which are mostly triple-negative breast cancer; (TNBC) lacking expression of estrogen receptors (ER), progesterone receptors (PR), and HER2 [2,3]. Breast cancer subtypes employ diverse mechanisms to evade therapies and suppress the immune response, prompting a shift in focus toward subtype-specific therapeutic approaches. Despite the success of immunotherapies in various cancers, including breast cancer, the effectiveness of immunotherapy in TNBC remains limited, yielding favorable outcomes only in selected patients [4]. In contrast to other subtypes, TNBC is associated with a high recurrence rate, poor prognosis, and low differentiation [1]. Patients with TNBC frequently experience relapse within five years post-surgery, resulting in unfavorable outcomes [4]. TNBC accounts for 10–20% of all breast cancer subtypes [5], and its resistance to endocrine and HER2-targeted therapies necessitates extensive exploration of its unique characteristics compared to other breast cancer subtypes [6]. Presently, ICIs have emerged as a promising strategy to stimulate the immune response and eliminate tumor cells within the tumor microenvironment (TME). Inhibiting checkpoints such as programmed death-1 (PD-1), programmed death ligand-1 (PD-L1), and cytotoxic T lymphocyte-associated antigen-4 (CTLA-4) regulates immunosuppressive cells, enhancing the required immune response in the TME of TNBC patients. PD-1/PD-L1 blockade is commonly employed in breast cancer trials, with FDA-approved monoclonal antibodies including Pembrolizumab, Nivolumab, Cemiplimab, Ipilimumab, Atezolizumab, Avelumab, and Durvalumab [7]. The immunotherapeutic landscape in breast cancer responds to the immunogenic nature of tumors, with TNBC exhibiting the highest immunogenicity among subtypes. Notably, Trastuzumab, a monoclonal antibody targeting HER-2 in HER2-positive tumors, has been effective in specific breast cancer subtypes [8]. Immunotherapeutic approaches in TNBC, especially in metastatic stages, have shown promising outcomes. Combining chemotherapy with Pembrolizumab (anti-PD-1) significantly improves overall survival rates in TNBC patients compared to chemotherapy alone [4]. Furthermore, the elevated expression of CTLA4 in TNBC patients presents an opportunity for targeting, although progress toward its therapeutic use is ongoing.

Additionally, the activation gene-3 (LAG3) and immunoglobulin and mucin domain protein 3 (TIM3) were highly found in Tumor-infiltrating lymphocytes (TILs) in TNBCs. Recent data showed that LAG3/TIM3 inhibitors modulate the clinical outcome of TNBC patients [9]. Of note, TNBC is categorized into 4 subtypes: immunomodulatory (IM), basal-like immune-suppressed (BLIS), mesenchymal-like (MES), and luminal androgen receptor (LAR). Importantly, IM subtypes showed the best response to immunotherapy compared to other subtypes, in which the TME was enriched with immunostimulatory and active immune cells. Consequently, researchers have undertaken an in-depth exploration of the TME in TNBC, aiming to discern optimal therapeutic strategies for each subtype [10]. This article highlights the primary challenges faced by Triple-Negative Breast Cancer (TNBC) in response to immunotherapies. Numerous ongoing trials are focused on addressing these challenges by modulating ICIs. Strategies include combining ICIs with other therapeutic agents and leveraging advanced technology for targeted delivery to the Tumor Microenvironments (TMEs). These approaches aim to overcome existing barriers and enhance the efficacy of immunotherapies in the context of TNBC.

## 2. Rationale Shift from Chemotherapy to Immunotherapy

Cancer Immunotherapy is an important milestone in the history of cancer treatment, reflecting our growing knowledge about the interaction between cancer cells and the body’s immune system. The traditional method of chemotherapy, which is one of the traditional cornerstones in cancer treatment, strikes the rapidly dividing cells regardless of whether they are cancerous or normal; the resultant side effects are often so severe that they leave them debilitated. Conversely, immunotherapy uses the host immune system to recognize and target the cancer cells without harming other cells. The basis of this strategy derives from the knowledge that cancerous cells utilize a variety of techniques to avoid recognition by the immune system. Thus, various methods of immunotherapy were developed such as ICIs including PD-L1 or CTLA-4, monoclonal antibodies, and adaptive cell therapies aiming to strengthen the immune system in recognizing and obliterating cancer [11]. Furthermore, with the rise of personalized medicine, and because tumors are inherently heterogeneous, personalized cancer treatments were developed to identify tumor-specific antigens, or by using vaccines to activate targeted immunologic responses towards cancers. Thus, durable responses were significantly associated with immunotherapy, compared to the gains from chemotherapy. However, some challenges remain such as finding biomarkers capable of predicting the tumor’s response to treatment, ensuring adequate safety measures against various toxicities related to the body’s reaction to treatment, and further improving the effectiveness of immunotherapy. further studies and clinical trials are still ongoing to improve this promising treatment.

## 3. Immunosuppressive Mechanisms within TME of TNBC Patient

In the intricate realm of Triple-Negative Breast Cancer (TNBC), understanding the dynamic interplay between cellular components within the Tumor Microenvironment (TME) is pivotal for unraveling the complexities of disease progression and treatment response. This exploration begins by scrutinizing the Cellular Components of the TME, where a consortium of cytokines, including Interleukin-10 (IL-10), IL-6, IL-4, IL-1β, IL-17, and Tumor Necrosis Factor-Alpha (TNF-α), orchestrates a multifaceted environment that collectively impacts immune cells, stromal elements, and tumor cells. These cytokines, with their intricate interactions, mold the immune landscape within the tumor and influence the efficacy of antitumor responses in TNBC patients. Beyond the cytokine-mediated processes, the narrative extends to Metabolic Reprogramming, where alterations in energy production and metabolite release shape an immunosuppressive TME. This metabolic shift not only influences immune cell functionality but also prompts a phenotypic transition in macrophages, further contributing to immunosuppression. The subsequent exploration delves into Changes in the configuration of the Extracellular Matrix, highlighting the paradoxical role of fibrosis in creating both physical barriers and immunosuppressive mechanisms within the TNBC TME. As therapeutic strategies targeting these intricate elements emerge, the collective insights foster a foundation for enhancing immunotherapy in TNBC to fortify the immune system’s ability to combat and control cancer cells.

### 3.1. Cellular Components of the TME

The TME of triple-negative breast cancer (TNBC) is a complex ecosystem composed of various cell types and extracellular components that interact with one another immunoregulatory cell subsets in the TME jointly construct an immunosuppressive network that weakens the antitumor effect of the host immune system (Figure 1) [12]. Among important cells in this ecosystem is the Tumor-Infiltrating Lymphocytes (TILs) [13]. These immune cells are known to infiltrate the tumor site. CD4+ T cells help coordinate immune responses, while CD8+ T cells are cytotoxic and directly attack cancer cells. The regulatory T cells (Tregs) are a subset of T cells that express the forkhead box P3 (Foxp3) transcription factor with immunosuppressive properties [12,14]. They maintain immune homeostasis but can be co-opted by the tumor to suppress antitumor immune responses in the TNBC [14,15]. In TNBC, TILs are often present but dysfunctional due to immune evasion mechanisms. The Myeloid-Derived Suppressor Cells (MDSCs) are a heterogeneous population of myeloid cells that are mainly involved with the suppression of the immune response by inhibiting T-cell activation and proliferation. Higher expression of MDSCs is a common feature noted in TNBCs compared to receptor-positive breast cancers associated with the activation of the chemokines CCL22 and CXCL2 at the tumor site causing significant metastasis cascades [16]. The Tumor-Associated Macrophages (TAMs) with their distinct polarization states of the antitumor pro-inflammatory M1 type and M2 type promote tissue repair and immunosuppression. In TNBC, there’s a shift toward M2-like TAMs, which contribute to immunosuppression by secreting different cytokines, chemokines, and growth factors [17]. M2 TAMs expressing CD163+ are associated with infiltration of stromal fibroblasts and mesenchymal transition state which were found to have aggressive phenotypes associated with poor survival rates among patients with TNBC [18].

### 3.2. Tumor-Induced Hypoxia

In addition to the conventional mechanisms of immunosuppressive responses at the tumor site due to activation of immune checkpoint pathways including the PD-1/PD-L1 Axis and CTLA-4 Pathway, there are other factors involved. For instance, hypoxia commonly happens in rapidly growing tumors that outsource the available blood supply, resulting in hypoxia within the TME. Hypoxia induces expresses the Angiogenic Growth Factors, which are associated with the stabilization and activation of hypoxia-inducible factors (HIFs) which are transcription factors that regulate the expression of genes involved in various aspects of tumor progression, including immunosuppression [19]. It can also promote the recruitment and expansion of myeloid-derived suppressor cells (MDSCs) and regulatory T cells (Tregs), both of which contribute to the immunosuppression [20]. In addition, hypoxia may upregulate the expression of PD-L1 in the hypoxic TME [21] and induce the production of exosomes and macrovesicles in breast cancer cells through HIF-dependent RAB22A expression, which can stimulate invasion and metastasis with poor overall survival rates [22].

### 3.3. Secretion of Immunosuppressive Cytokines

Cytokines are small signaling proteins secreted by various cell types, including tumor cells, immune cells, and stromal cells, within the TME. These cytokines play a crucial role in modulating the immune response and influencing tumor progression. Different cytokines contribute to the suppression of immune response in the tumor microenvironment. These cytokines are secreted by cancer cells and stromal cells in the tumor microenvironment including immune cells, endothelial cells, and fibroblasts. For example, cytokines secreted by cancer-associated fibroblasts (CAFs) cause immunosuppression, tumor cell proliferation, and remodeling of the ECM. immunoregulatory cell subsets in the TME jointly construct an immunosuppressive network that weakens the antitumor effect of the host immune system in TNBC, several cytokines are involved in creating an immunosuppressive microenvironment such as transforming growth factor-beta (TGF-β) and interleukin-10 (IL-10).

TGF-β is a multifunctional cytokine with immunosuppressive properties. In the TNBC TME, TGF-β is often overexpressed and released by both tumor cells and immune cells [23]. It has several immunosuppressive including the inhibition of T lymphocytes proliferation and activation of cytotoxic CD8+ T cells and NK cells, reducing their ability to target and kill tumor cells [24]. In addition, it promotes the differentiation and activation of regulatory T cells (Tregs), which have immunosuppressive functions and dampen antitumor immune responses. TGF-β also modulates the phenotype of tumor-associated macrophages (TAMs) toward the immunosuppressive M2-like phenotype [25]. IL-10 has immunosuppressive properties, dampening the immune response by inhibiting the production of pro-inflammatory cytokines and downregulating the expression of MHC class II molecules on APCs.

Interleukin-10 (IL-10): IL-10 is an anti-inflammatory cytokine that is often elevated in the TNBC TME. Its immunosuppressive effects include suppressing the production of pro-inflammatory cytokines, such as interferon-gamma (IFN-γ) and tumor necrosis factor-alpha (TNF-α), by immune cells [26]. In addition, it inhibits the expression of major histocompatibility complex (MHC) molecules on antigen-presenting cells (APCs), reducing their ability to present tumor antigens to T cells [27].

Interleukin-6 (IL-6): IL-6 is a pro-inflammatory cytokine that plays a dual role in cancer. In TNBC, it can have both pro- and anti-tumorigenic effects; IL-6 can promote the recruitment and activation of immune cells to the TME, potentially enhancing antitumor immune responses. However, IL-6 can also contribute to immunosuppression by inducing the differentiation of Tregs and promoting the expansion of MDSCs [28].

Interleukin-4 (IL-4): IL-4 is an immunoregulatory cytokine that can contribute to immunosuppression in TNBC by promoting the polarization of macrophages toward the M2-like phenotype, which supports tumor growth and inhibits immune responses [29].

Interleukin-1β (IL-1β): IL-1β is a pro-inflammatory cytokine produced by various cells within the TME, including tumor cells, immune cells, and stromal cells. In TNBC, IL-1β can promote tumor growth and immunosuppression by Inducing the production of other pro-inflammatory cytokines and chemokines, thereby contributing to a chronic inflammatory environment and regulating the differentiation and activity of immune cells, potentially promoting a more immunosuppressive phenotype [30].

Interleukin-17 (IL-17): IL-17 is a cytokine primarily produced by a subset of T cells known as Th17 cells. In the TNBC TME, IL-17 can have dual roles It can stimulate the recruitment of immune cells to the TME, potentially enhancing antitumor immune responses [31]. However, chronic IL-17 signaling may also contribute to tumor progression by promoting angiogenesis and tissue remodeling [32].

Tumor Necrosis Factor-Alpha (TNF-α): TNF-α is a pro-inflammatory cytokine that can have both pro- and anti-tumor effects. In TNBC, it can promote tumor cell death and inhibit angiogenesis. However, chronic TNF-α signaling can also contribute to the recruitment of immunosuppressive cells like MDSCs and TAMs [33].

These cytokines collectively contribute to the dynamic and multifaceted nature of the TNBC TME. Their intricate interactions with immune cells, stromal cells, and tumor cells shape the immune landscape within the tumor, ultimately impacting the progression and response to treatment in TNBC patients. Understanding these cytokine-mediated processes is crucial for developing targeted therapies that can modulate the TME and enhance antitumor immune responses in TNBC.

### 3.4. Metabolic Reprogramming

Metabolic reprogramming, a hallmark of cancer cells, involves a shift in energy production and utilization to sustain the high proliferation rates observed in tumors [34]. In TNBC, this metabolic switch often entails a preference for aerobic glycolysis, known as the Warburg effect [35]. Beyond providing energy, these metabolic alterations play a pivotal role in establishing an immunosuppressive TME. One key aspect of metabolic reprogramming in TNBC is the increased production of metabolites that contribute to immunosuppression. For instance, lactate, a byproduct of glycolysis, is abundantly produced by cancer cells and has been identified as a potent immunosuppressive factor [36]. High levels of lactate in the TME create an acidic environment that impairs the function of immune cells, including T cells and natural killer cells, thereby attenuating the antitumor immune response.

Moreover, altered amino acid metabolism in TNBC contributes to the generation of immunosuppressive signals [37]. Cancer cells often upregulate enzymes involved in tryptophan metabolism, leading to the production of kynurenine. Elevated kynurenine levels have been linked to the inhibition of T cell function and the promotion of regulatory T cell (Treg) differentiation, both of which contribute to immune evasion in the TME [37].

Additionally, metabolic reprogramming influences the composition and function of immune cells within the TME. Cancer cells can induce a phenotypic shift in macrophages, promoting the polarization of tumor-associated macrophages (TAMs) toward an M2-like immunosuppressive phenotype. This shift is orchestrated by metabolites, such as lactate and adenosine, which are abundantly produced in the glycolytic TME of TNBC [36,38]. Adenosine is a purine nucleoside that can suppress T cell function by binding to specific adenosine receptors (A2A and A2B) on the surface of T cells. When adenosine binds to these receptors, it inhibits T cell activation, cytokine production, and cytotoxicity [38]. TAMs, in their immunosuppressive state, further contribute to the suppression of antitumor immunity.

Understanding the intricate connections between metabolic reprogramming and immunosuppression in TNBC has prompted the exploration of novel therapeutic strategies. Targeting key metabolic pathways, such as glycolysis and amino acid metabolism, holds promise for disrupting the immunosuppressive TME. Inhibitors of glycolytic enzymes and immune checkpoint molecules are currently under investigation as potential combination therapies to enhance the effectiveness of immunotherapy in TNBC [39].

### 3.5. Changes in the Configuration of the Extracellular Matrix in the TME in TNBC

One prominent feature of the tumor microenvironment (TME) in TNBC is the development of fibrosis, which has recently garnered attention for its potential role in shaping immunosuppressive mechanisms. Fibrosis, the excessive deposition of extracellular matrix components, is commonly associated with chronic inflammation and tissue remodeling [40]. In the context of TNBC, fibrosis contributes to the formation of a dense and fibrotic TME, creating a barrier that can impede effective drug delivery and immune cell infiltration [41]. Paradoxically, this fibrotic response also plays a crucial role in the establishment of immunosuppressive mechanisms within the TME. The fibrotic TME in TNBC is characterized by the activation of cancer-associated fibroblasts (CAFs) and the increased deposition of proteins such as collagen and fibronectin. These changes create a physically and biochemically hostile environment, limiting the infiltration and activity of Tumor-infiltrating lymphocytes (TILs) [42]. This physical barrier, combined with the secretion of immunosuppressive cytokines by CAFs, hampers the efficacy of the immune response against cancer cells.

Understanding the role of fibrosis in immunosuppression within the TME has led to novel therapeutic strategies aimed at targeting fibrotic elements to enhance immunotherapy in TNBC. Antifibrotic agents, such as losartan and pirfenidone, have shown promise in preclinical studies by reducing collagen deposition and improving TIL infiltration [43]. By mitigating the fibrotic barrier, these agents may enhance the effectiveness of immunotherapeutic approaches, such as ICIs, which aim to unleash the immune system’s ability to recognize and attack cancer cells.

## 4. Overcoming Mechanisms of Immune Resistance within TME of TNBC Patients

### 4.1. Monotherapy-ICIs

Immune checkpoints, which are expressed on the surface of immune cells (IC) and tumor cells (TC), serve as a communication bridge between cells in the tumor immune microenvironment (TME). Immunogenic status within the TME reflects the communication status between cells. Recently, the blockade of immune checkpoints has been used to remodel the immune response in the TME. Hence, an ICI is a monoclonal antibody that targets a specific immune checkpoint, in which PD-1 is highly expressed on T, NK, and myeloid immune cells while PD-L1 is found either on tumor or activated macrophages, T cells, and CAFs [1,44]. Notably, within the context of TME, the activation of cytotoxic immune cells is achieved independently through the action of PD-1. Conversely, when PD-1 binds with its ligand PD-L1, it inhibits the proliferation of T and B cells by downregulating the PI3K/AKT pathway through the presence-activated SH2 protein tyrosine phosphatase 2 (SHP2). Moreover, this process expedites cell death and enables tumor cells to evade elimination [1].

The United States Food and Drug Administration (FDA) granted approval in the year 2019 for the utilization of the PD-L1 inhibitor (Atezolizumab) in conjunction with chemotherapy (Abraxane) as a treatment for patients with metastatic TNBC [45]. Later, numerous cancer studies have been involved in using ICIs as alternative cancer therapy, particularly in triple-negative breast cancer.

The initial stage of the KEYNOTE-012 clinical research investigation involved a cohort of 111 patients diagnosed with triple-negative breast cancer (TNBC). These patients received a PD-L1 inhibitor, specifically Pembrolizumab, administered intravenously at a dose of 10 mg/kg every two weeks. The overall response rate (ORR) at this stage was determined to be 18.5%. Subsequently, the subsequent phase of the trial, where PD-1 inhibitors were administered at a dosage of 10 mg/kg intravenously every three weeks, resulted in a reduced ORR of 5.7%. Notably, no significant alterations in either the ORR or survival rates were observed during the third and final phase III trials. It is pertinent to highlight that single ICIs have demonstrated only modest responses in the early stages, as elucidated by Zhu et al. [46].

Phase 1b JAVELIN solid tumor investigation employed a PD-L1 inhibitor known as Avelumab to manage patients with metastatic breast cancer, including 58 cases of triple-negative breast cancers (TNBCs). The Avelumab treatment resulted in a 5.3% objective response rate (ORR) in TNBCs. The ORR observed in TNBC patients who exhibited positivity for PD-L1 was determined to be 22.2%, whereas the ORR in PD-L1-negative patients was found to be 2.6% [47]. Conversely, Avelumab displayed a lower ORR of 3% when considering the overall population of individuals with metastatic breast cancer. Importantly, the safety profile of Avelumab is adaptable, and its potential for achieving a cure is limited to specific subtypes of metastatic breast cancer. Consequently, numerous clinical investigations are currently underway to explore the combination of Avelumab with novel treatment strategies to enhance the favorable outcomes associated with the monotherapy [47].

PD-L1 is prominently exhibited in tumor cells and its interaction with PD1-immune cells is employed as a defensive mechanism against predominantly T cells. Furthermore, the notable expression of PD-L1 on malignant cells is linked to the dissemination of cancer cells and unfavorable prognosis in lymph nodes. As a result, a fresh immunotherapy strategy is being devised to explore the inherent impact of PD-L1 in TNBC without PD-1. In laboratory conditions, the investigation employed PD-L1-siRNA to mute the PD-L1 [48] and assess the cell proliferation, cell migration [49], tumor apoptosis, and T cell induction to TMIE. Significantly, the inhibition of PD-L1 demonstrated noteworthy findings when tested in conjunction with T cells. Silencing the expression of PD-L1 in a breast cancer cell line (MDA-231 cells) yields interesting discoveries. The reduction in PD-L1 expression upon gene silencing results in a significant decrease in the migration of tumor cells towards the wound-healing assay. Additionally, there is an elevation in the production of pro-inflammatory cytokines, namely TNF-α, IL-2, and IFN-γ, by NK and T cells. Conversely, there is a decrease in the production of anti-inflammatory cytokines, such as IL-10 and TGF-β, by Tregs cells. These novel findings emphasize the potential of PD-L1 silencing as a promising therapeutic approach in the treatment of TNBC [48].

### 4.2. Dual ICIs

PD-1/PD-L1 inhibition during the initial phases in patients with triple-negative breast cancer resulted in a response rate ranging from 18% to 24% [50]. As of 2023, the coadministration of Pembrolizumab, a monoclonal antibody targeting the programmed cell death protein 1 (PD-1), and Doxorubicin in individuals with anthracycline-naïve metastatic triple-negative breast cancer (mTNBC) has displayed a synergistic effect in stimulating the cellular immune response and managing the disease. It is noteworthy that a significant 67% of mTNBC patients exhibited an overall response to the treatment, accompanied by a notable activation of T cells [51]. At present, the blockade of PD-1/PD-L1 remains insufficient in the context of triple-negative breast cancer (TNBC), partially attributed to the existence of transforming growth factor-beta (TGF-β) as indicated by Yin et al. [52].

Using either anti-PD-1 or anti-CTLA4 as monotherapy has demonstrated a limited immune response against tumors. However, the combination of anti-PD-1 and anti-CTLA4 in the animal model of TNBCs enhances the effectiveness of anti-PD-1 and anti-CTLA4 against TNBC. Specifically, the administration of anti-CTLA4 (Ipilimumab and Tremelimumab) results in an expansion of the T cell population and a significant decrease in T regs cells, leading to complete recovery from TNBCs in 80% of mice [53].

Remarkably, there is a significant expression of lymphocyte activation gene-3 (LAG-3) and programmed cell death protein-1 (PD-1) in the lymphocytes that infiltrate the tumor. This phenomenon plays a crucial role in enabling tumor cells to evade encountering the immune cells within the tumor microenvironment (TME). Furthermore, LAG-3 is also expressed in natural killer cells, B cells, and dendritic cells [54].

An in vivo study used BALB/c mice as a TNBC model to investigate the synergistic effect of the dual blockade on both LAG3 and PD-1. Significantly, LAG3 and PD1 blockade had achieved a pronounced reduction (*p* < 0.05) in tumor’s weight and growth in TNBC BALB/c mice compared with a single blockade of LAG3 or PD-1 which showed less reduction in tumor weight [9]. Presently, a clinical study has explored the inhibitory effects of LAG-3 and PD-1 within the tumor microenvironment of metastatic TNBCs. Despite the enrollment of 37 patients in this study, LAG-3 inhibitors have effectively overcome the resistance observed with PD-1/PD-L1 inhibitors in TNBCs [54].

The roster of immune checkpoints encompasses an additional constituent, the V-set immunoregulatory receptor (also known as VISTA), which is identified as a crucial immune checkpoint that is influential in the management of various malignancies, particularly triple-negative breast cancer (TNBC) [55]. VISTA functions as a receptor on T cells and as a ligand on antigen-presenting cells (APCs). Data that has been reported indicates that 80% of immune cells and 18% of tumor cells express VISTA in a group of 254 patients who were in the early stages of TNBC and had not received any treatment. 

Initially, the administration of anti-VISTA alone had a positive impact on tumors by reducing the presence of MDSCs in the tumor microenvironment (TME). Subsequently, a combination therapy involving Cycloheximide, radiotherapy, and the dual blockade of PD1 and VISTA was utilized. This combined therapy resulted in an extended overall survival in tumors, achieved through an increase in the presence of CD8+ T-cells that infiltrate the tumor and a decrease in the presence of MDSCs [56]. Indeed, these findings ensure the promised outcome will be picked back upon using multi-faceted ICIs.

### 4.3. Chemotherapy Combined ICIs

Striking success has been achieved in cancer outcomes and immune resistance via combining ICIs with specific chemotherapy. Particularly, platinum-based chemotherapy promoted ICI efficacy by making tumor cells more sensitive to PD1/PD-L1 inhibitors and highly expressed to PD-L1 [57]. Recent Phase III clinical trials conducted on patients diagnosed with triple-negative breast cancer have provided evidence to support the effectiveness of combining chemotherapy with immunotherapy, specifically ICIs as a potential treatment option [58]. A randomized Phase III trial, known as KEYNOTE-522, was undertaken to assess the safety, efficacy, and pathological complete response (pCR) of neoadjuvant Pembrolizumab-chemotherapy in previously untreated TNBC patients. The study involved two distinct groups, whereby one group received Pembrolizumab-chemotherapy while the other group received placebo-chemotherapy, in which the pCR was 64.8% and 51.2% respectively, thus reaffirming the efficacy of Pembrolizumab in enhancing the therapeutic outcomes of chemotherapy in Phase III TNBCs including PD-L1 subgroups [59]. Further recent and consistent randomized Phase III trial, known as KEYNOTE-355, successfully enrolled a subset of 87 Japanese patients with PD-L1. Within this trial, 61 patients were treated with Pembrolizumab-chemotherapy, while the remaining 26 patients received placebo-chemotherapy. The results indicated a significant improvement in the 18-month overall survival rate and the 12-month progression-free survival for the group receiving Pembrolizumab-chemotherapy, compared to the placebo-chemotherapy group [60]. However, in contrast, Phase III Mpassion131 revealed that the combination of Atezolizumab and Paclitaxel was unsuccessful in enhancing significant progression survival in patients with positive PD-L1 advanced TNBC, as compared to the use of Paclitaxel alone [61].

New strategies are currently focused on the inhibition of angiogenesis within the tumor microenvironment (TME), which creates a barrier against the infiltration of CD8+ T cells into the TME. Despite the collaborative efforts of immune cells to disrupt tumor vessels, particularly through the activity of IFNG-secreting CD8+ T cells and M2 macrophages, tumor angiogenesis still impacts the infiltration of CD8+ T cells. Recent preclinical and in vivo studies have emphasized the incorporation of the angiogenesis inhibitor Famitinib with the anti-PD1 drug Camrelizumab. These studies have shown that the objective response rate among 48 advanced immunomodulatory triple-negative breast cancer patients was 81.3%. Additionally, CD8+ T cells accounted for more than 10% of the patient population, and the median progression-free survival was 13.6 months (95% CI, 8.4–18.8). Interestingly, an objective response was observed in all patients with PD-L1 positive tumors and in 69% of patients with PD-L1 negative tumors. Furthermore, the combination therapy was found to be most beneficial for patients with both CD8+ and PD-L1 positive tumors [62].

### 4.4. Cancer Vaccine Combined ICIs

The utilization of CTLA4 blockade remains restricted and the prognosis of cancer can be controlled in specific subtypes of triple-negative breast cancer (TNBC). Interestingly, recent data suggests that the inclusion of a CTLA4 inhibitor in conjunction with a MUC1 mRNA nanovaccine enhances the therapeutic impact of the anti-CTLA-4 monoclonal antibody and adjusts the immune resistance within the tumor immune microenvironment (TME) [63]. Both preclinical and clinical investigations ensure the remarkable nature of a therapeutic combination. Thus far, the immune-modifying function of a CTLA4 inhibitor combined with an MUC1 mRNA nanovaccine in the tumor microenvironment (TME) of triple-negative breast cancer (TNBC) has been evaluated by examining the cytotoxic immune cells and cytokines. It is worth noting that the combination of anti-CTLA4 and MUC1 mRNA vaccine resulted in an increased presence of cytotoxic CD8+ T cells and elevated levels of IL12 and IFN gamma. Furthermore, this combination treatment significantly reduced the levels of Treg, MDSC, TNF-α, IL-6, and TGF-β within the TME. Notably, the combined therapy also led to a decrease in STAT3 and phosphor-STAT3 levels. STAT3 is a critical transcription factor involved in cancer migration and invasion, and it is commonly utilized as a biomarker for poor cancer prognosis [63]. Furthermore, this incorporation leads to a noteworthy augmentation in tumorous cells undergoing apoptosis.

### 4.5. Combining ICIs with TME Metabolites

In actuality, the tumor microenvironment is primarily characterized by a multitude of metabolic disturbances. The presence of cancer cells is associated with a state of accelerated metabolism, whereby metabolites play a pivotal role in facilitating tumor cell proliferation, migration, and maturation. An illustrative example of this phenomenon involves the non-essential amino acid, glutamine, which serves as a valuable nutritional resource for tumor cells. Predominantly, tumor cells consume glutamine, thereby leaving a limited supply available for immune cells. Consequently, altering the metabolic profile of glutamine in tumor cells indirectly stimulates the immune response by reducing the presence of PD-L1 tumor cells. A recent investigation has documented that the inhibition of the glutamine transporter, specifically the carrier family 7 member 5 (SLC7A5), in conjunction with anti-PD-L1 treatment, significantly augments the immune response within the tumor microenvironment [64]. The inhibition of SLC7A5 leads to the enhancement of CD4+ and CD8+ T lymphocytes, thereby boosting the immune response while concurrently diminishing tumor progression and cellular migration.

Profoundly, overexpression of immune checkpoints in tumor cells is accompanied by an increase in metabolite production which provokes immune suppression such as lactate and glycolysis. Further studies focused on investigating tumor metabolites in TNBC cell lines and the impact of their interaction with immune checkpoints, particularly PD-L1.

Interestingly, several metabolites engage in interactions with immune checkpoints, including COX-2, TGF-β, and choline kinase-α (Chk-α), the latter being significantly upregulated in tumor cells. This upregulation predominantly correlates with the evasion of immune response by tumors [65]. An inverse correlation was observed between PD-L1 and Chk-α, whereby the inhibition of Chk-α leads to a significant upregulation of PD-L1. This upregulation of PD-L1 enhances immune suppression by altering the metabolic profile of tumor cells, specifically affecting metabolites such as glutamine, glutamate, and lactate. These metabolites are known to induce immune resistance in tumors. Consequently, modulating the levels of tumor metabolites holds promise in enhancing immune response through the utilization of ICIs [65]. It is observed that the administration of any therapeutic intervention that indirectly reduces the expression of Chk-α will lead to an increase in the expression of PD-L1. However, the simultaneous reduction in the expression levels of both Chk-α and PD-L1 will counteract this effect.

### 4.6. Cytokines–IFN/TGF Beta Crosstalk with ICIs

Breast cancer cells often use immune checkpoint blockade to evade the immune system’s response, as they have high levels of PD-L1 which helps them create anti-immune responses. To enhance immune response, blocking immune checkpoints is the best approach, with dual blockade being the most rational for TNBC. The regulation of PD-L1 expression can be achieved at the transcriptional or post-transcriptional level, and IFN-γ serves as a crucial regulator for PD-L1 in the tumor microenvironment (TME) of TNBC. It is worth noting that IFN-γ exhibits the remarkable ability to impede tumorigenesis by inducing apoptosis in tumor cells and inhibiting angiogenesis. However, it can also upregulate immune checkpoints in the TME, thereby facilitating the evasion of tumor cells from the immune system. Recent evidence indicates the presence of UBR5, a critical transcription factor for the IFN-γ signaling pathway, in the TME of TNBC. UBR5 induces the expression of PD-L1, and the dual blockade of UBR5 and PD-L1 may increase the survival rate and overcome the resistance of tumor cells [66].

The use of ICIs in treating triple-negative breast cancer (TNBC) remains ineffective due to multiple factors, for instance, the presence of Transforming Growth Factor (TGF-β), which is known to reprogram the tumor microenvironment in Triple Negative Breast Cancer. It suppresses the immune system by increasing collagen production in cancer-associated fibroblasts. Furthermore, TGF-β interrupts both macrophage polarization and tumor-infiltrating lymphocyte activity, making it a highly adaptable environment in the progression of tumors. It is worth noting that the dual therapy approach, which entails blocking both TGF-β and PD-1 or PD-L1, has been proven to be highly effective in the treatment of TNBC in mice models [67]. The anti-tumor effect was more pronounced when using PD-1-TGF-β blockade compared to PD-L1-TGF-β blockade. PD-1-TGF-β blockade in TNBC’s TME effectively decreases collagen deposition while simultaneously augmenting CD8+ cells and tumor-infiltrating lymphocytes. These findings offer a remarkably optimistic dual treatment solution for TNBC and warrant serious consideration as a feasible treatment alternative [67].

### 4.7. Anti CD25/Anti CD47 Combined with ICIs

The imbalance of immune cells within the TME’s TNBC drew the researcher’s interest in revealing the most prominent cell types within all stages of TNBC and understanding their impacts on TME. Consequently, they have discovered that the most prevalent cell types in primary TNBC patients are CD25^high^ effector regulatory T cells (eTregs), while effector T cells (Cytotoxic CD8+) are significantly diminished. Furthermore, the abundance of eTregs is linked to resistance against anti-PD-1 therapy in TNBC. Hence, the utilization of synergistic therapy involving anti-CD25 and anti-PD-1 has fostered an immune response within the TME, thereby enhancing the efficacy of anti-PD-1 treatment in TNBC. Notably, this synergistic therapy can reverse the cell ratio and improve the immune response. Data have shown that 90% of tumors are eliminated by the combination of anti-CD25 and anti-PD1, resulting in a high ratio of CD8+ T cells to low eTregs CD25^high^ [68].

Tumor cells exhibit diverse surface proteins, and the integral membrane protein CD47 is found to be amplified in tumor cells. Consequently, the investigation of triple-negative breast cancer (TNBC) elucidates the significant contribution of CD47 in the development and advancement of tumor load. Furthermore, the blockade of CD4 has demonstrated an enhanced immunogenicity immune response against the tumor. While the employment of monotherapy, either anti-PD-L1 or anti-CD47, has facilitated the immune response, the incorporation of anti-CD47 with anti-PD-L1 has conspicuously diminished the tumor burden through the augmentation of cytotoxic T cells that secrete intratumoral granzyme B [69].

### 4.8. Chemokine Inhibitors Cross-Talk with ICIs

Genomic analysis has revealed that TNBCs TME exhibits elevated expression of CXCR4 and its ligand, SDF-1 (also known as CXCL12). Both primary TNBCs and their metastatic forms display high levels of the G-protein-coupled receptor CXCR4. The CXCL12/CXCR4 axis attracts immune suppressive cells to TME, including M2-phenotype macrophages, regulatory T cells (Tregs), and myeloid-derived suppressor cells (MDSCs). Additional studies have demonstrated the critical role of the CXCL12/CXCR4 axis in promoting cancer cell proliferation and invasion, as well as its importance in tumor prognosis. Recently, researchers have explored combining this axis with ICIs to enhance the efficacy of treatment. In 2021, a research study focused on disrupting CXCL12/CXCR4 to eliminate immune suppression in TME and improve treatment outcomes of ICIs in TNBC. CXCR4 antagonist is used in the form of encapsulated liposome to encounter Plerixafor, an FDA-approved drug, which is another name for CXCR4. Intracellular and extracellular CXCR4 can be effectively targeted using liposomal treatment with AMD3100, resulting in a long-lasting response. Notably, combining anti-PD1 with liposomal AMD3100 has been shown to promote an effective immune response against tumor cells. Moreover, the use of chemokine cross-talk ICIs has been demonstrated by Lu, Qiu, and Su in 2021 to eliminate ICI resistance in TNBC [70].

### 4.9. Gut Microbiota Crosstalk with ICIs

Earlier investigations showed the gut microbiota as a critical player in most human diseases. Hence, host gut microbiota cross-talks various physiological processes via modulating immune response, epithelial production, and metabolite production to keep hemostasis. Interestingly, in cancer diseases, human gut microbiota work as immunosuppressive and oncogenic actors based on the abundance of microbiota type and other factors [71]. Furthermore, the presence of gut microbiota has been detected within the tumor microenvironment (TME) and is encountered in most cancer treatments. Therefore, the production of microbiota-specific metabolites stimulated immunotherapy among patients diagnosed with melanoma. An extensive investigation in 2020 scrutinized more than seven tumor tissues, which ultimately detected distinct microbiota profiles within tumor tissue when compared to normal tissue [72]. Particularly, breast tumor tissues exhibited a substantial abundance of diverse microbiotas. The alterations in gut microbiota diversity primarily occur in relation to the various stages and subtypes of cancer [72]. Accordingly, specific microbiota, namely *Cloacibacterium*, *Blastomonas*, *Stakelama*, *Filibacter*, *Anaerostipes*, *Alloprevotella*, and PRD01a011B, have been identified in HER2-positive breast cancer patients, as opposed to HER2-negative tumors.

Hence, the examination conducted by Wang and colleagues (2022) has primarily concentrated on comprehending the tumor microenvironment (TME) of triple-negative breast cancer (TNBC) patients. Within this context, the researchers have observed the prevalence of *Clostridiales* and metabolite trimethylamine N-oxide (TMAO) within the TME of TNBC. TMAO and *Clostridiales* have exhibited a positive correlation with TNBC patients undergoing immunotherapy [73]. To elaborate, TMAO can elicit gasdermin E (GSDME)-mediated pyroptosis in tumor cells. Therefore, tumor cell death is elicited by the activation of gasdermin and caspase 3. Additionally, it stimulates a greater CD8+ anti-cell mediated immunity response within the tumor microenvironment (TME) of triple-negative breast cancer (TNBC) [74].

In this regard, TAMO and choline-rich diet might be used as potential therapy in TNBC. Patients with enriched TMAO plasma had a strong immune response to anti-PDI inhibitors and increased survival rates. Upon these findings, further studies continue to investigate the microbiota’s contribution to TNBC therapy. Further research found the efficacy of PDL-I inhibitors in the TNBC mice model was synergistically improved by involving either Western/Mediterranean diet intake and gut microbiota. In which the diet intake and gut *Akkermansia muciniphila* significantly improved the reaction to PD-L1 blockade by up to 70–40% [74]. The list of treatment regimens used in TNBC are shown below in (Table 1).

## 5. Ongoing Challenges of Immunotherapy in TNBC Patients

Despite extensive investigations into understanding the Tumor Microenvironment (TME) of Triple-Negative Breast Cancer (TNBC) and promising clinical trials, several challenges persist. Tumor studies have shown that ICIs exhibit better responses in cases with a high tumor mutational burden (TMB) and favorable immune infiltrate disposition (FID). Tumors characterized by high TMB and FID are termed “hot tumors”, capable of eliciting a robust immune response to ICIs, as observed in melanoma, liver, and kidney cancers. Notably, TNBC stands out among breast cancers for having a high TMB, especially in TNBC and Her-2 enriched tumors with T Infiltrate Lymphocytes (TILs), indicating heightened immunogenicity and a more favorable ICI response [7]. However, multiple challenges arise for TNBC patients undergoing ICI treatment. Tumor heterogeneity, known for its molecular and genetic diversity, presents a challenge in predicting responses to immunotherapy due to variable immune cell infiltration and PD-L1 expression. Unlike some cancers, TNBC lacks a definitive biomarker for predicting responsiveness to immunotherapy, with PD-L1 expression not universally correlating with therapeutic outcomes. The effectiveness of ICIs in TNBC is hampered by emerging resistance mechanisms, including alterations in the tumor microenvironment and upregulation of alternative immune checkpoints.

Despite showing success in early-stage TNBC, ICIs exhibit limited efficacy in metastatic settings due to the aggressive nature of metastasis within the immunosuppressive microenvironment. Immune-related adverse events, such as gastritis, colitis, pneumonitis, and thyroid problems, pose risks due to autoimmune reactions, requiring prompt management.

The ongoing research focuses on improving ICI efficacy in TNBC through combination approaches involving chemotherapy, targeted therapy, and immunotherapy. Identifying optimal combinations and treatment sequences remains an active area of research.

## Figures and Tables

**Figure 1 biomedicines-12-00369-f001:**
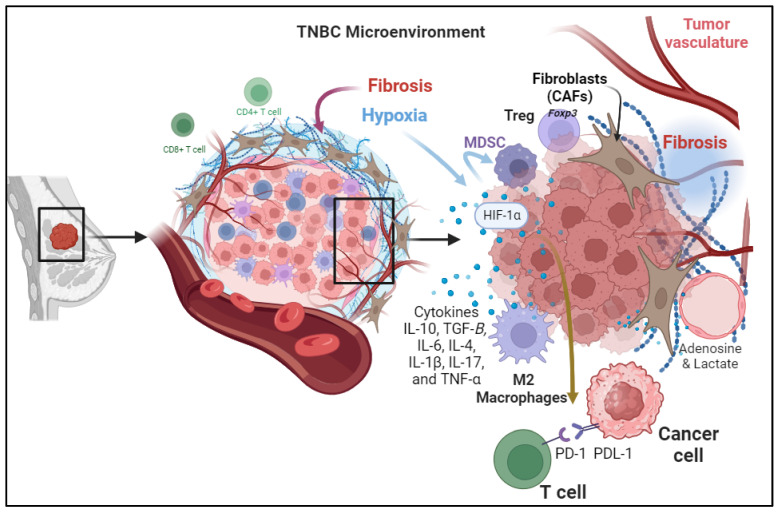
The intricate components contributing to the Tumor Microenvironment (TME) in Triple-Negative Breast Cancer (TNBC) are depicted. Cellular Components of the TME, highlighting the diverse cell types such as Tumor-Infiltrating Lymphocytes (TILs), Myeloid-Derived Suppressor Cells (MDSCs), and Tumor-Associated Macrophages (TAMs) and their roles in immunosuppression. Tumor-induced hypoxia, illustrating the impact of hypoxia on immune response through various pathways including immune checkpoint activation and myeloid cell recruitment. Secretion of Immunosuppressive Cytokines, outlining the role of cytokines, such as TGF-β, IL-10, IL-6, IL-4, IL-1β, IL-17, and TNF-α, in shaping the immunosuppressive microenvironment. Metabolic Reprogramming, showcases the shift in energy metabolism and the production of immunosuppressive metabolites such as adenosine and lactic acid, influencing immune cell function. Changes in the configuration of the Extracellular Matrix (ECM) emphasize the role of fibrosis in creating a barrier that hinders effective drug delivery and immune cell infiltration. Understanding these components is vital for developing targeted therapies to modulate the TME and enhance antitumor immune responses in TNBC patients.

**Table 1 biomedicines-12-00369-t001:** List of treatment regimens used to overcome resistance to immune checkpoint in Triple-Negative Breast Cancer patients.

Therapeutic Approach	Potential Mechanisms and Outcomes	References
PD-L1 blockade (pembrolizumab) PD-1—blockade (pembrolizumab)	ORR 18.5% ORR 5.7%, modest immune response	[7]
PD-L1-siRNA	Increase TNF-α, IL-2, IFN-γ decrease IL-10 and TGF-β,	[48]
CTLA4/PD-1blockade	Increase T cell population decrease T regs cells	[53]
VISTA blockade (Clone 13F3) VISTA blockade+ Cycloheximide+ radiotherapy,	decrease MDSCs in TME Increase CD8+ T-cells+ decrease MDSCs	[56]
LAG3/PD-1 blockade	decrease tumor weight & size	[54]
Chemotherapy -ICIs (Pembrolizumab)	pCR up to 64.8% increase in Pembrolizumab-group/51.2% in placebo group	[60]
Angiogenesis/PD-1 blockade	ORR up to 81.3% in TNBC CD8+ T cells > 10%	[62]
CTLA4 blockade & MUC1 mRNA vaccine	Increased cytotoxic CD8+ T cells Increase IL-12 & IFNγ decrease Tregs, TNF-α, IL-6, TGF-β, STAT3 and phospho STAT3 levels.	[63]
UBR5/PD-L1 blockade	Increase PD-L1 & Increase immune suppression Increase survival rate	[66]
PD-1-TGF-β blockade	decrease collagen deposition Increase CD8+ cells & Increase TILs	[67]
CD25- PD-1 blockade	Increase CD8+ T cells & decrease CD25^hig^	[68]
CD47- PD-L1 blockade	Increase cytotoxic T cells	[69]
anti-PD1 with liposomal AMD3100 Plerixafor,	Promote an effective immune response	[69]
Gut Microbiota (*Akkermansia muciniphila*) & PD-L1 blockade	Increase efficacy of PD-L1 blockade by up to 70–40%	[74]

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
