# Peer review of "Breaking Barriers: The Promise and Challenges of Immune Checkpoint Inhibitors in Triple-Negative Breast Cancer"

_biomedicines, 2024, doi:10.3390/biomedicines12020369_

Round 1

Reviewer 1 Report

Comments and Suggestions for Authors

The manuscript simply describes the mechanisms of immuno-resistance of cancer cells in TNBC and provides some approved immunotherapies and future perspectives to overcome these challenges. 

Immunotherapy in TNBC is a recent medical success and impressive results from phase III trials were worth mentioning.  

Nevertheless, this narrative article does not present the actual on-going studies or future perspectives in a meaningful and clinical sense of utility.

Author Response

Response: Thank you for your constructive feedback. We have added details about phase III trials in line 390.

Reviewer 2 Report

Comments and Suggestions for Authors

The topic is current and interesting. TNBC is an evolving therapeutic area. The absence of endocrine targets and Her-2 targets necessitates extensive exploration of other target in order to adding some other therapeutic options to chemotherapy.  

The argument is not so original, but your review is well written and exhaustive.

In the section 4. Overcoming Mechanisms of Immune …” (lines 284 -563) suggest to adding some tables for summarize the complexity of the information reported and for being more fluent the review.

Author Response

Response: Thank you for your constructive feedback. We have added a table that summarizes the information in line 593.

Reviewer 3 Report

Comments and Suggestions for Authors

The manuscript presents a concise and competent review of therapeutic modalities, including immune checkpoint inhibitors targeting PD-1, PD-L1, and CTLA-4, explored in preclinical and clinical trials. The review is comprehensive, discussing the rationale for the shift from chemotherapy to immunotherapy, immunosuppressive mechanisms within the microenvironment of  triple-negative breast cancer, overcoming mechanisms of immune resistance within tumor microenvironment, including the crosstalk of gut microbiota with immune checkpoint inhibitors, and ongoing challenges of immunotherapy in patients with triple-negative breast cancer.

Minor remarks:

Line 96: “Patient” or “Patients”?

Line 262: “extracellular rather than “extra cellular”

Line 556: “choline-rich diet”?

Latin names should be in italics

Lines 258-261: Please provide citation(s)

References should be formated according to the requirements of the journal.

Comments on the Quality of English Language

Rare spelling and grammar errors

Author Response

Minor remarks:

Line 96: “Patient” or “Patients”?

Response: Thank you for your constructive feedback. Correction Done

Line 262: “extracellular rather than “extra cellular”

Response: Thank you for your constructive feedback. Correction Done

Line 556: “choline-rich diet”?

Response: Thank you for your constructive feedback. Choline is a water-soluble nutrient essential for human life. Gut microbial metabolism of choline results in the production of trimethylamine (TMA), which, upon absorption by the host is converted into trimethylamine-N-oxide (TMAO) in the liver.

Latin names should be in italics.

Response: Thank you for your constructive feedback. Correction Done

Lines 258-261: Please provide citation(s)

Response: Thank you for your constructive feedback. The following citation is added” Ren X, Cheng Z, He J, Yao X, Liu Y, Cai K, Li M, Hu Y, Luo Z. Inhibition of glycolysis-driven immunosuppression with a nano-assembly enhances response to immune checkpoint blockade therapy in triple negative breast cancer. Nat Commun. 2023 Nov 2;14(1):7021. doi: 10.1038/s41467-023-42883-2. PMID: 37919262; PMCID: PMC10622423.

References should be formated according to the requirements of the journal.

Response: Thank you for your constructive feedback. Correction Done